# Human Papillomavirus Oncoproteins Confer Sensitivity to Cisplatin by Interfering with Epidermal Growth Factor Receptor Nuclear Trafficking Related to More Favorable Clinical Survival Outcomes in Non-Small Cell Lung Cancer

**DOI:** 10.3390/cancers14215333

**Published:** 2022-10-29

**Authors:** Jinn-Li Wang, Wei-Jiunn Lee, Chia-Lang Fang, Han-Lin Hsu, Bo-Jung Chen, Hsingjin-Eugene Liu

**Affiliations:** 1Division of Hematology and Oncology, Department of Pediatrics, Wan Fang Hospital, Taipei Medical University, Taipei 11031, Taiwan; 2Department of Pediatrics, School of Medicine, College of Medicine, Taipei Medical University, Taipei 11031, Taiwan; 3Department of Urology, School of Medicine, College of Medicine, Taipei Medical University, Taipei 11031, Taiwan; 4Department of Medical Education and Research, Wan Fang Hospital, Taipei Medical University, Taipei 11031, Taiwan; 5Cancer Center, Wan Fang Hospital, Taipei Medical University, Taipei 11031, Taiwan; 6Department of Pathology, School of Medicine, College of Medicine, Taipei Medical University, Taipei 11031, Taiwan; 7Department of Pathology, Taipei Medical University Hospital, Taipei Medical University, Taipei 11031, Taiwan; 8Division of Pulmonary Medicine, Department of Internal Medicine, Wan Fang Hospital, Taipei Medical University, Taipei 11031, Taiwan; 9Department of Pathology, Shuang-Ho Hospital, Taipei Medical University, New Taipei City 23561, Taiwan; 10Graduate Institute of Clinical Medicine, College of Medicine, Taipei Medical University, Taipei 11031, Taiwan

**Keywords:** human papillomavirus, epidermal growth factor receptor, survival, cisplatin, lung cancer

## Abstract

**Simple Summary:**

Lung cancer is the leading cause of cancer death in the world. Identifying prognostic factors is crucial to improve the survival time of those with lung cancer. Our previous studies have reported that human papillomavirus (HPV) infections and epidermal growth factor receptor (EGFR) expression are associated with a better survival prognosis in lung adenocarcinoma. The purpose of this study was to detect the molecular evidence of HPV oncoproteins interfering with EGFR nuclear trafficking related to better prognosis in lung cancer. Based on the study results for a better response to cisplatin in transfected HPV 16E5/16E6/16E7 H292 xenograft animal models, as well as better survival in lung adenocarcinoma patients with either 16E6/18E6 or EGFR expression, we suggest that clinicians should adjust the treatment protocol according to HPV 16E6/18E6 expression and EGFR expression to increase the overall survival time in lung cancer.

**Abstract:**

High-risk human papillomavirus (HPV) infections and epidermal growth factor receptor (EGFR) expression have been reported to be associated with more favorable survival outcomes in lung adenocarcinoma patients. In this study, we utilized transfected HPV 16E5/16E6/16E7 H292 cells to investigate the mechanism of HPV oncoproteins interfering with EGFR nuclear trafficking related to a better response to cisplatin. Furthermore, we correlated HPV 16E6/18E6 expression and differentially localized EGFR expression with the clinical association and survival impact in lung adenocarcinoma patients. Our results found significantly higher phosphorylated nuclear EGFR expression upon epidermal growth factor stimulus and better responses to cisplatin in transfected HPV 16E5/16E6/16E7 NCI-H292 cells and xenograft animal models. Our data were compatible with clinical results of a high correlation of HPV 16E6/18E6 and EGFR expression in non-small cell lung cancer tissues and the synergistic effects of both with the best survival prognosis in a lung adenocarcinoma cohort, especially in patients with older age, no brain metastasis, smoking history, and wild-type EGFR status. Cumulatively, our study supports HPV 16E5/16E6/16E7 oncoproteins interfering with EGFR nuclear trafficking, resulting in increased sensitivity to cisplatin. HPV 16E6/18E6 and EGFR expression serve as good prognostic factors in lung adenocarcinoma patients.

## 1. Introduction

Lung cancer remains the leading cause of cancer death worldwide according to 2020 global cancer statistics [1]. Therefore, the identification of its causes is essential for the prevention and treatment of its deadly disease. In addition to smoking, high-risk human papillomavirus (HPV) infections have been a recognized risk factor in multiple nongenital cancers, including lung cancer [2,3,4]. We have previously reported that HPV infections result in a more favorable prognosis in Taiwanese lung adenocarcinoma patients [5]. However, the underlying reason why HPV infections confer a better prognosis remains unclear.

The epidermal growth factor receptor (EGFR) signaling pathway plays a critical role in tumorigenesis and tumor progression [6]. EGFR overexpression is frequently found in lung cancer tissues and plays a key role in promoting lung cancer propagation [7,8]. In addition to the classic EGFR signaling pathway, EGFR nuclear trafficking is involved in cancer progression through different cellular processes [9,10]. Our previous study reported that lung adenocarcinoma with different localizations of EGFR proteins responds favorably to platinum-based chemotherapy [11]. Therefore, the effect of EGFR nuclear trafficking on lung carcinogenesis should not be neglected.

The HPV E5 oncoprotein has been shown to induce keratinocyte proliferation through the upregulation of the EGFR signaling pathway [12]. The activation of EGFR signaling by HPV E5 leads to the activation of several proto-oncogenes and finally results in cell proliferation and transformation [12]. HPV E6 and E7 oncoproteins have been addressed to play roles in HPV E5-induced tumorigenesis and increased EGFR mRNA levels in human keratinocytes [13,14]. Based on these observations, we propose that HPV immortalizes cancer cells, not only by inhibiting tumor suppressors (p53 or Rb) [15,16] but also by activating the EGFR signaling pathway, eventually resulting in increased proliferation. However, the association between HPV oncoproteins and EGFR nuclear translocation is not clear. 

Since cervical cancer cells are more sensitive to cisplatin and high-risk HPV infections cause more than 90% of cervical cancer [17,18], HPV oncoproteins might contribute to a better response to cisplatin. The EGFR nuclear translocation in lung carcinogenesis was associated with better prognosis in our previous study; therefore we reason that HPV oncoproteins may interfere with EGFR nuclear trafficking and may change lung cancer behavior related to more sensitivity to cisplatin. To prove our hypothesis, we established the transfected HPV 16E5/16E6/16E7 NCI H292 (lung mucoepidermal carcinoma cells) cells and xenograft animal models to explore the molecular mechanisms through which HPV oncoproteins interfere with EGFR nuclear trafficking. Furthermore, we correlated HPV 16E6/18E6 expression and EGFR (membranous and nuclear) expression with the clinical association and their survival impact in a non-small cell lung cancer (NSCLC) cohort.

## 2. Materials and Methods

### 2.1. Cell Culture, Plasmids, and Transfection

The human lung mucoepidermoid carcinoma cell line H292 (NCI-H292, BCRC 60372), purchased from the Bioresource Collection and Research Center (BCRC, Taiwan), was maintained in the Roswell Park Memorial Institute medium (Gibco, ThermoFisher Scientific, Waltham, MA, USA) supplemented with 10% heat-inactivated fetal bovine serum, 100 units/mL penicillin, and 100 mg/mL streptomycin. Cells were grown at 37 °C in a humidified atmosphere containing 5% CO_2_. MSCV-N 16E5 (HPV 16E5), MSCV-N 16E6 (HPV 16E6), and MSCV-N 16E7 (HPV 16E7) plasmids were obtained from Addegene (Watertown, MA, USA), the nonprofit plasmid repository. The fusion protein constructs were then transfected into the NCI-H292 cell line, as per the instructions, through electroporation (Amaxa^®^ Cell Line Nucleofector^®^ Kit T, Lonza, Basel, Switzerland). Puromycin (0.6 μg/mL, Protech Technology Enterprise Co., Ltd., Taipei, Taiwan) was used to select single cell lines (H292-HPV16E5, H292-HPV16E6, and H292-HPV16E7 cells). The transfection efficiency was determined using the tagged protein, hemagglutinin (HA, ThermoFisher Scientific).

### 2.2. Flow Cytometry

We collected 1 × 10^6^ cells, centrifuged them at 1000 rpm for 4 min, and removed the supernatant. We then added 100 μL of phosphate-buffered saline (1× PBS, pH 7.4) per tube containing 3% bovine serum albumin and added 5 μL of antibody (anti-human EGFR Antibody Biolegend #352904, San Diego, CA, USA) to each tube. Cells were incubated for 30 min on ice in the dark, washed three times, centrifuged at 400 *g* for 5 min, resuspended in ice-cold PBS, and put on ice in the dark during transport. They were incubated for 30 min on ice and analyzed in a flow cytometer (Invitrogen Attune™ NxT Acoustic Focusing Cytometer, Waltham, MA, USA).

### 2.3. Kinetic Study of Nuclear Localization of EGFR

H292, H292-HPV16E5, H292-HPV16E6, and H292-HPV16E7 cells were stimulated with 100 ng/mL epidermal growth factor (EGF, Sigma-Aldrich, St. Louis, MO, USA). They were incubated at 37 °C for 30 min. Then, nuclear and non-nuclear fractions were isolated and subjected to Western blotting with anti-EGFR (sc-03, Santa Cruz, Dallas, TX, USA) and anti-pEGFR (sc-12351R, Santa Cruz) antibodies on nitrocellulose membranes. The same membranes were also probed with histone 3 (Histone H3 Antibody #9715, Cell Signaling Technology, for the nuclear fraction) or tubulin (α-Tubulin Antibody #2144, Cell Signaling Technology (Danvers, MA, USA), for the non-nuclear fraction) as a loading control.

### 2.4. MTT Assay

Due to the clinical evidence that HPV-positive lung cancer patients respond to platinum-based chemotherapy, the 3-(4,5-dimethylthiazol-2-yl)-2,5-diphenyltetrazolium bromide (MTT) assay was used to detect cell viability after cisplatin administration. Adhesion cells (5 × 10^3^ cells; 100 μL) were seeded into the wells of a 96-well plate containing various concentrations of cisplatin (2, 4, 6, 8, 10, and 12 μg/mL). After incubation for 72 h at 37 °C, 10 μL of the MTT solution was added to each well, and the plates were incubated for a further 3 h at 37 °C. Absorbance was measured at 550 nm using a 96-well reader (Gen-5). Each experiment was performed in triplicate wells for each drug concentration.

### 2.5. Animal Studies

Finally, we established the xenograft animal models using the H292, H292-HPV16E5, H292-HPV16E6, and H292-HPV16E7 cells to determine the treatment responses after cisplatin administration. BALB/c nude mice were purchased from BioLASCO Taiwan Co., Ltd. (Taipei, Taiwan). All mice were maintained in laminar-flow cabinets under specific pathogen-free conditions at room temperature in a 24 h night–day cycle. To establish the xenograft models, 5-week-old BALB/c nude mice were anesthetized with pentobarbital; then, H292-HPV16E5, H292-HPV16E6, and H292-HPV16E7 cells (5 × 10^5^) were resuspended in a 1:1 mixture of PBS and GFR-Matrigel and injected subcutaneously into the right flanks of mice using a 27-gauge needle. After 7 days, these mice were intraperitoneally administered 10 mg/kg cisplatin or saline once a week for 3 weeks. To determine the drug effects, the body weight of mice and tumor volume (width^2^ × length/2) were monitored every 3 days. After 28 days, the tumor weight and final body weight were recorded after sacrifice. All animal experimental protocols were approved by the Institutional Animal Care and Use Committee of Taipei Medical University.

### 2.6. Patient Population

The same study cohort described in our previous papers was enrolled with 243 paraffin-embedded lung cancer tissue blocks between 2008 and 2014 retrieved from Wan Fang Hospital, Taipei Medical University [5,11]. Clinical characteristics including age, gender, smoking history, histology, tumor stage, EGFR mutations, and the last follow-up date were obtained from patient medical records. This study was approved by the Joint Institutional Review Board of Taipei Medical University.

### 2.7. Immunohistochemistry

HPV E4 is responsible for viral replication and might disappear later during the late stages of cervical carcinogenesis [19,20]; therefore, we detected HPV16E6/18E6 expression to determine HPV infections due to HPV-induced oncogenesis. Since fresh tissues are needed in the in situ hybridization of HPV RNA or DNA to prevent false-negative findings, we used immunostain to detect HPV 16E6/18E6 expression in lung cancer tissue blocks. The methods for nuclear EGFR (**n**EGFR), membranous EGFR (**m**EGFR), and HPV 16E6/18E6 immunostaining were as mentioned in our previous studies [5,11]. The detailed methodology of IHC is available upon request in Appendix A. All IHC-stained specimens were graded by two pathologists (C.L. Fang and B.J. Chen) blindly. IHC scoring was graded as high expression (mild or strong) and low expression (negative or weak) according to international IHC scoring in previous studies [21,22]. The images of **n**EGFR and **m**EGFR expression were shown in our previous paper [11]. The image of HPV 16E6/18E6 expression is shown in Figure 1.

### 2.8. Statistical Analyses

The EGFR protein levels in different transfected H292-HPV16E5, H292-HPV16E6, and H292-HPV16E7 cells were determined using a *t-*test for dual comparisons. The χ^2^ test was used to determine the relationship between HPV 16E6/18E6 expression and differentially located EGFR expressions as well as the clinical characteristics of these biomarkers. To determine the synergistic effect of HPV 16E6/18E6 expression and EGFR expression, we divided the data into three subgroups: **E6^−^tEGFR^−^, E6^+^tEGFR^−^ or E6^−^tEGFR^+^, and E6^+^tEGFR^+^** (**tEGFR** is defined as either high **m**EGFR or **n**EGFR expression in our previous paper) [11]. The Kapla–-Meier estimate using the log-rank test was employed to evaluate the survival outcome in the three subgroups. Overall survival (OS) was defined as the time between the date of diagnosis and that of death from any causes or the date of censorship (date of final follow-up). Using Cox proportional hazards models, we evaluated the hazard ratio (HR) and corresponding confidence interval (CI) to identify potential prognostic factors. All statistical methods were two-sided, and significance was accepted at *p* < 0.05. SAS version 9.4 was used for all statistical analyses.

## 3. Results

### 3.1. High EGFR Protein Expression in Transfected H292-HPV16E5, H292-HPV16E6, and H292-HPV16E7 Cells

We designed the basic experiments to provide molecular evidence of the relationship between HPV oncoproteins and EGFR overexpression. First, we confirmed the transfection efficiency of HPV16E5, HPV16E6, and HPV16E7 in NCI-H292 cells (Figure 1A). Then, we examined the difference in EGFR expression in transfected H292-HPV16E5, H292-HPV16E6, and H292-HPV16E7 cells through Western blotting and flow cytometry. Compared with untransfected H292 cells, transfected H292-HPV16E5, H292-HPV16E6, and H292-HPV16E7 cells showed a significantly high EGFR expression (*t-*test, *p* < 0.01, *p* < 0.01 and *p* < 0.001, respectively, Figure 2A,B).

### 3.2. Increased Phosphorylated Nuclear EGFR Protein Levels after EGF Stimulus in Transfected H292-HPV16E5, H292-HPV16E6, and H292-HPV16E7 Cells

We examined the difference in nuclear EGFR protein levels in transfected H292-HPV16E5, H292-HPV16E6, and H292-HPV16E7 cells after EGF stimulus. We found evidence of EGFR translocation after EGF (100 ng/mL) stimulus in untransfected NCI-H292 cells (Appendix A). Thus, we investigated the differences in EGFR protein expression in transfected H292-HPV16E5 H292-HPV16E6, and H292-HPV16E7 cells after EGF (100 ng/mL) stimulus for 15 min in comparison with the expression in H292 cells and we found higher nuclear phosphorylated EGFR protein levels in transfected H292-HPV16E5 and H292-HPV16E6 cells (see Figure 2C)

### 3.3. Better Treatment Responses to Cisplatin in Transfected H292-HPV16E5, H292-HPV16E6, and H292-HPV16E7 H292 Cells

Finally, we established the xenograft animal models using transfected H292-HPV16E5, H292-HPV16E6, H292-HPV16E7, and H292 cells to determine the treatment responses to cisplatin. In the MTT assay, we found no differences in the viability of transfected H292-HPV16E5, H292-HPV16E6, and H292-HPV16E7 cells compared with that of H292 cells (Figure 3A). Furthermore, no significant difference was found in cisplatin toxicity, with no significant mice body weight change after tumor grafting between mice administered saline and cisplatin (Figure 3B). Then, we investigated the tumor volume per 3 days after tumor xenografting and found significantly reduced tumor volume in xenograft animals established using transfected H292-HPV16E5, H292-HPV16E6, and H292-HPV16E7 cells (*p* < 0.05, *p* < 0.05 and *p* < 0.01, respectively, see Figure 3C–E). Furthermore, the tumor inhibition percentage was significantly higher in the animal models with transfected H292-HPV16E5, H292-HPV16E6, and H292-HPV16E7 cells (59.9%, 62.1%, and 67.4%, respectively, Figure 3F).

### 3.4. High HPV 16E6/18E6 Expression Related to High Nuclear and Membranous EGFR Expression in 243 Primary Lung Cancer Tissues

We used the IHC scores of the three biomarkers (HPV 16E6/18E6 and **n**EGFR/**m**EGFR expression) to explore the association between HPV infections and differentially located EGFR expression. The χ^2^ test results revealed significantly positive correlations of HPV 16E6/18E6 expression with **n**EGFR as well as **m**EGFR expression in 243 NSCLC tissues (*p* < 0.0001 and *p* < 0.0001, respectively). The data are shown in Table 1.

### 3.5. Lower Prevalence of E6^+^tEGFR^+^ Expression in Lung Adenocarcinoma Patients at an Advanced Stage

Regarding the different biology of lung squamous and adenocarcinoma, we focused the analyses on the clinical characteristics of 173 lung adenocarcinoma patients. We found a significantly lower prevalence of any expression (**E6^+^tEGFR^−^/E6^−^tEGFR^+^**) or both expressions (**E6^+^tEGFR^+^**) in lung adenocarcinoma patients with a higher nodal stage (**E6^+^tEGFR^−^/E6^−^tEGFR^+^** vs. **E6^−^tEGFR^−^,** OR [95% CI] = 0.46 [0.22–0.97], *p* = 0.040; **E6^+^tEGFR^+^** vs. **E6^−^tEGFR^−^,** OR [95% CI] = 0.38 [0.17–0.85], *p* = 0.018, respectively). Furthermore, as shown in Table 2, lung adenocarcinoma patients with the distant TNM stage had a significantly lower prevalence of both HPV 16E6/18E6 and EGFR expressions (**E6^+^tEGFR^+^** vs. **E6^−^tEGFR^−^,** OR [95% CI] = 0.39 [0.16–0.94], *p* = 0.034). However, other categorical variables, such as age, gender, smoking history, and EGFR mutations, were not associated with HPV 16E6/18E6 expression or EGFR expressions. 

### 3.6. Lung Adenocarcinoma Patients with E6^+^tEGFR^+^ Expression Had the Longest Survival Time with a Better Treatment Response to Cisplatin

Next, we analyzed the clinical effects of the combination of HPV 16E6/18 E6 expression and EGFR expressions in 173 lung adenocarcinoma patients. In analyses stratified by HPV 16E6/18E6 and EGFR expression (**E6^−^tEGFR^−^**, **E6^+^tEGFR^−^ or E6^−^tEGFR^+^**, and **E6^+^tEGFR^+^**), we found that lung adenocarcinoma patients with **E6^+^tEGFR^+^** had the longest survival time (overall *p* = 0.055, median = 66 months; **E6^+^tEGFR**^+^ vs. **E6^−^tEGFR^−^**, HR [95% CI] = 0.77 [0.62−0.96], *p* = 0.019). Specifically, the clinical significance of **E6^+^tEGFR^+^** expression was identified in lung adenocarcinoma patients with older age, no brain metastasis, smoking history, and wild-type EGFR status. Figure 4 shows the survival curve and detailed information. To examine the treatment response of HPV oncoprotein and EGFR expression in lung adenocarcinoma patients, we found a significantly decreased hazard risk in patients with any HPV 16E6/18E6 or EGFR expression in older age, no brain metastasis, smoking history, and wild-type EGFR status after cisplatin (HR [95% CI] = 0.35 [0.13–0.96], 0.42 [0.19–0.91], 0.27 [0.09–0.84] and 0.38 [0.15–0.96], respectively. Table 3). However, no similar findings were determined in patients after tyrosine kinase inhibitor (TKI, gefitinib, or erlotinib) and radiation (Appendix A).

## 4. Discussion

This study supported the hypothesis that HPV oncoproteins interfered with EGFR nuclear trafficking resulting in EGFR overexpression and conferred chemosensitivity to cisplatin in transfected HPV 16E5/16E6/16E7 H292 cells and the xenograft animal models. These results were consistent with the clinical findings of a high correlation between HPV 16E6/18E6 expression and different localization of EGFR expression in lung cancer tissues and lung adenocarcinoma patients with both expressions with the longest survival time. Subgroup analyses showed that patients with both expressions after cisplatin treatment had a significantly lower hazard risk. Together, our findings indicate that HPV oncoproteins might change the behavior of lung cancer cells through changes in EGFR nuclear trafficking, and the expression of HPV 16E6/18E6 and EGFR serves as a predictive biomarker of survival benefits in lung adenocarcinoma patients.

This study has provided evidence of HPV oncoproteins interfering with EGFR nuclear trafficking through increased EGFR expression and increased phosphorylated nuclear EGFR expression after EGF stimulus in transfected HPV 16E5/16E6/16E7 H292 cell lines. In cervical cancer cells, the HPV early gene product, HPV 16E5, modulates the ligand-dependent activation of EGFR in EGFR overexpression [23,24]. Furthermore, HPV E6/E7 oncoproteins in combination with EGF promote cervical cancer cell proliferation [25], and HPV E6 causes prolonged tyrosine receptor signaling, resulting in the internalization of phosphorylated receptors, such as EGFR protein [26,27]. Taken together, HPV oncoproteins might have effects such as stimulating factors, resulting in increased EGFR nuclear translocation.

In addition, better responses to cisplatin have been identified in the xenograft (using transfected HPV 16E5/16E6/16E7 H292 cells) animal models. The results provided molecular evidence in our prior studies showing better responses to platinum-based chemotherapy in lung adenocarcinoma patients with HPV infections and EGFR expression [5,11]. With the E7-mediated degradation of AMBRA1 (a regulator of autophagy), HPV sensitizes oropharyngeal squamous cells to cisplatin-induced apoptosis by inhibiting autophagy [28]. Furthermore, the HPV E7 oncoprotein induces S phase entry along with the DNA damage response, and it enhances apoptosis and the cisplatin effect [29]. HPV early genes may act as the stimulating factor to enhance EGFR nuclear trafficking, and nuclear EGFR translocation modulates DNA repair following cisplatin administration, further resulting in cancer cell apoptosis [30]. Taken together, the results indicate that HPV-infected lung cancer cells are more sensitive to cisplatin.

Consistent with the results of basic experiments with HPV oncoproteins enhancing EGFR nuclear translocation, we found a strong correlation between HPV 16E6/18E6 expression and **n**EGFR/**m**EGFR expression in NSCLC tissues with a preference in early-stage lung adenocarcinoma. The relationship between both biomarkers in lung cancer has been less reported. Nevertheless, inconsistent findings in head and neck cancers have been obtained regarding the correlation, with a strong association found between HPV infections and EGFR expression in laryngeal squamous cell carcinoma [31], and a strong correlation with phosphorylated EGFR expression in HPV-negative tonsillar squamous cell carcinoma [32]. The prognostic value of HPV infection and EGFR expression in head and neck cancers has been reported as HPV infections on the survival outcomes are identified as a better prognostic factor [33,34], and phosphorylated EGFR is associated with longer survival and as a potential therapeutic biomarker [35]. Similarly, our study has addressed the synergistic effect of HPV 16E6/18E6 and EGFR expression on the survival outcome in a lung adenocarcinoma cohort. Since a continued investigation is recommended in the treatment pathway of HPV-positive head and neck cancer, detailed clinical information is needed to prove the effects of combined EGFR expression and HPV infections on clinical outcomes in lung cancer.

However, this study has several limitations. First, we limited our analyses to apoptosis alternation in transfected HPV 16E5/E6/E7 H292 cells compared with H292 cells after cisplatin administration. Second, this is a single-institution retrospective study with small sample size. Third, we did not establish a patient-derived (with HPV infections) xenograft model due to limited tissue availability to investigate the molecular mechanism for better treatment responses to platinum-based chemotherapy.

In the future, we will develop patient-derived (using fresh lung adenocarcinoma cancer tissues with or without HPV infections) xenograft models to test the treatment response after cisplatin administration. The tumor mass in the xenograft models will be examined for apoptosis alternation such as caspase enzyme activity or DNA repair enzymes (such as Uracil-DNA glycosylase) and tested for the interaction of HPV oncoprotein and EGFR nuclear trafficking-related factors such as cyclin D1. The detailed molecular mechanism of lung adenocarcinoma patients with HPV infections will be determined.

## 5. Conclusions

Through a series of basic experiments, we have provided evidence of HPV oncoproteins interfering with the EGFR nuclear trafficking pathway related to better treatment response to cisplatin. Therefore, this study suggests that personalized medicine should be included in the treatment protocol for lung cancer patients according to the status of HPV oncoproteins and EGFR expression.

## Figures and Tables

**Figure 1 cancers-14-05333-f001:**
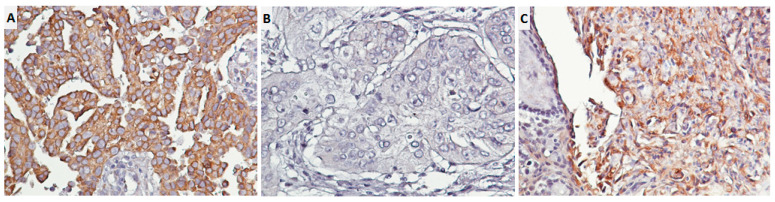
Immunostaining for HPV 16E6/18E6 proteins in lung adenocarcinoma tissues. (**A**) High expression; (**B**) low expression; (**C**) positive control (cervical squamous carcinoma); 400×.

**Figure 2 cancers-14-05333-f002:**
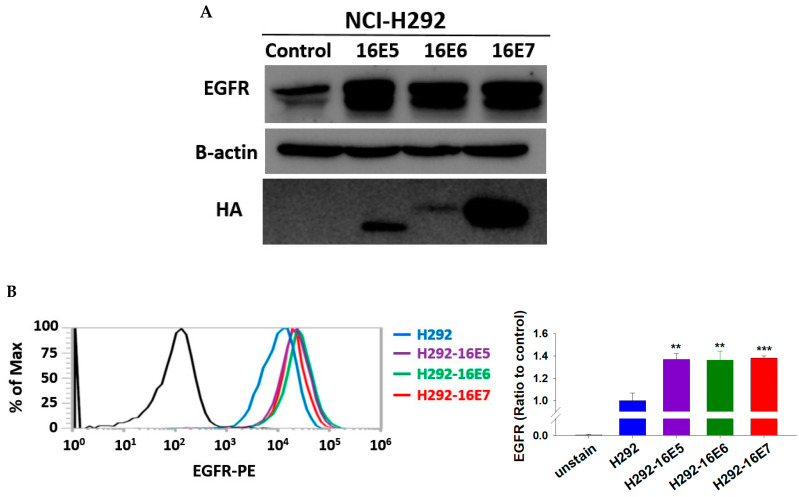
Epidermal growth factor receptor (EGFR) expression and EGFR nuclear translocation in transfected H292-HPV16E5, H292-HPV16E6, and H292-HPV16E7 cells. (**A**) Immunoblotting to assess the efficiency of transfection and EGFR expression. Hemagglutinin (HA) was used to evaluate the transfection efficiency. B-actin was used as internal control. (**B**) Flow cytometry to assess differences in EGFR protein expression. ** *p* < 0.01; *** *p* < 0.001. (**C**) Increased phosphorylated EGFR nuclear and nonnuclear proteins after EGF 100 ng/mL stimulus for 15 min in transfected H292-HPV16E5 and H292-HPV16E6 cells. Tubulin and histone 3 were used as the internal control of nonnuclear and nuclear proteins, respectively. The uncropped blots are shown in Appendix A.

**Figure 3 cancers-14-05333-f003:**
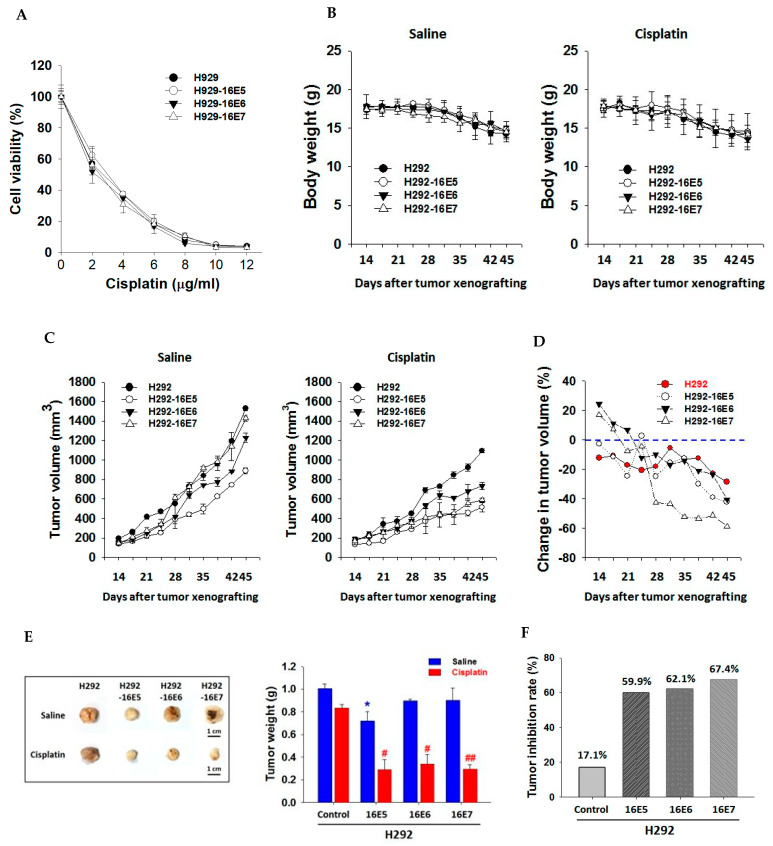
Better treatment response in H292-HPV16E5, H292-HPV16E6, and H292-HPV16E7 cells. (**A**) Cell viability after cisplatin administration. Viability was determined using the MTT assay. (**B**) Body weight of nude mice after tumor xenografting. (**C**) Tumor volume after tumor xenografting. (**D**) Change in tumor volume after tumor xenografting. (**E**) Photographs of tumor size and comparison of tumor weight. * *p* < 0.05; **^#^** *p* < 0.05; **^##^** *p* < 0.01. (**F**). Tumor inhibition rate.

**Figure 4 cancers-14-05333-f004:**
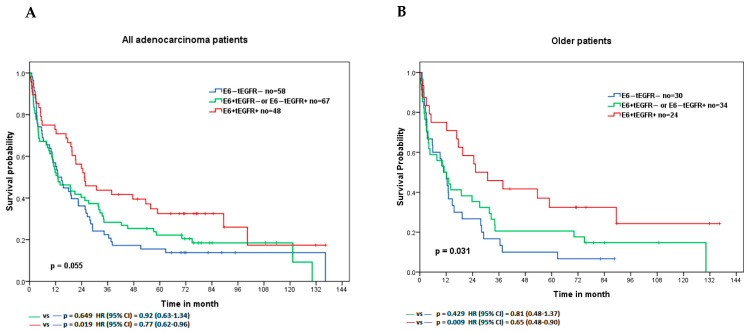
Survival analyses of lung adenocarcinoma patients. (**A**) For all patients; (**B**) for patients with older age; (**C**) for patients without brain metastasis; (**D**) for patients with smoking history; (**E**) for patients with wild-type epidermal growth factor receptor (EGFR) status.

**Table 1 cancers-14-05333-t001:** Correlation of HPV 16E6/18E6 expression and differentially located EGFR expression in 243 non-small-cell lung cancer tissues.

Parameter		HPV 16E6/18E6 Expression		*p*-Value
		Low	High	
Nuclear EGFR	Low	121 (79.1) ^a^	42 (46.7)	**<0.0001 ***
	High	32 (20.9)	48 (53.3)	
Membranous EGFR	Low	106 (69.3)	35 (38.9)	**<0.0001 ***
	High	47 (30.7)	55 (61.1)	

χ^2^ test, ^a^ The number in parenthesis is the percentage of the column. ***** Statistical significance.

**Table 2 cancers-14-05333-t002:** Distribution of HPV 16E6/18E6 expression related to total EGFR expression in 173 lung adenocarcinoma patients.

Characteristics	E6^−^tEGFR^−^ (N = 58)	E6^+^tEGFR^−^ or E6^−^tEGFR^+^ (N = 67)	E6^+^tEGFR^+^ (N = 48)	*p*-Value
Age (median = 71 yr)				0.984
Younger	28 (48.28) ^a^	33 (49.25)	24 (50.00)	
Older > 70 yr	30 (51.72)	34 (50.75)	24 (50.00)	
Gender				0.811
Female	35 (60.34)	39 (58.21)	26 (54.17)	
Male	23 (39.66)	28 (41.79)	22 (45.83)	
Smoking				0.767
Never	39 (67.24)	43 (64.18)	29 (60.42)	
Current or past	19 (32.76)	24 (35.82)	19 (39.58)	
Tumor stage (2 missing)				0.239
T1/T2	24 (42.10)	37 (56.06)	21 (43.75)	
T3/T4	33 (57.90)	29 (43.94)	27 (56.25)	
Nodal stage (1 missing)				**0.039**
L0/L1	16 (27.59)	30 (45.45) ^b^	24 (50.00) ^c^	
L2/L3	42 (72.41)	36 (54.55)	24 (50.00)	
Metastasis				0.155
without	18 (31.03)	30 (44.78)	23 (47.92)	
with	40 (68.97)	37 (55.22)	25 (52.08)	
TNM stage				0.084
Localized (stage I/II)	11 (18.97)	16 (23.88) ^d^	18 (37.50) ^e^	
Distant (stage III/IV)	47 (81.03)	51 (76.12)	30 (62.50)	
Brain metastasis				0.191
without	45 (77.59)	42 (62.69)	34 (70.83)	
with	13 (22.41)	25 (37.31)	14 (29.17)	
EGFR mutations ^f^ (20 missing)				0.362
Wildtype	29 (54.72)	32 (54.24)	17 (41.46)	
Mutationse	24 (45.28)	27 (45.76)	24 (58.53)	

χ^2^ test, Note: Boldfaces as statistical significance. ^a^ The number in parentheses is the percentage of the column. ^b^ OR = 0.46 (0.22–0.97) *p* = 0.040, ^c^ OR = 0.38 (0.17–0.85) *p* = 0.018 L0/L1 as reference; ^d^ OR = 0.75 (0.31–1.77) *p* = 0.505, ^e^ OR = 0.39 (0.16–0.94) *p* = 0.034 Localized as reference, ^f^ Including 1 in exon 18, 37 in exon 19, 3 in exon 20, 33 in exon 21 and 1 in exon 19/20.

**Table 3 cancers-14-05333-t003:** Comparison of treatment response in 173 lung adenocarcinoma patients after platinum-based chemotherapy.

Parameters	No	Median (m)	HR (95% CI)	*p*-Value
Total				
E6^+^/tEGFR^+^	36	31.4	0.58 (0.32–1.04)	0.066
E6^−^tEGFR^−^	20	20.8	1	
Older patients				
E6^+^/tEGFR^+^	18	31.7	0.35 (0.13–0.96)	**0.042 ***
E6^−^tEGFR^−^	8	16.1	1	
No brain metastasis				
E6^+^/tEGFR^+^	20	57.2	0.42 (0.19–0.91)	**0.028 ***
E6^−^tEGFR^−^	14	20.8	1	
Smokers				
E6^+^/tEGFR^+^	12	44.9	0.27 (0.09–0.84)	**0.024 ***
E6^−^tEGFR^−^	6	17.7	1	
Wildtype EGFR				
E6^+^/tEGFR^+^	12	29.0	0.38 (0.15–0.96)	**0.041 ***
E6^−^tEGFR^−^	9	18.0	1	

Cox proportional model, **E6^+^/tEGFR^+^** as **E6^+^tEGFR^+^**, **E6^+^tEGFR^−^** or **E6^−^tEGFR^+^**. ***** Statistical significance.

## Data Availability

The data generated in this study are available within the article and its Appendix A from the corresponding author upon request.

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
