# Peer review of "Human Papillomavirus Oncoproteins Confer Sensitivity to Cisplatin by Interfering with Epidermal Growth Factor Receptor Nuclear Trafficking Related to More Favorable Clinical Survival Outcomes in Non-Small Cell Lung Cancer"

_cancers, 2022, doi:10.3390/cancers14215333_

Round 1

Reviewer 1 Report

The results presented in this study by Wang et al regarding the role of HPV oncoproteins in conferring platinum sensitivity in lung non-small cell carcinoma is interesting. However, there are certain issues that need to be addressed:

1. The authors need to provide the detailed immunohistochemistry protocol in the supplement.

2. The authors report the findings on HPV E16E6 and HPV18E6 protein expression. Although E6 is the major HPV oncoprotein, HOV E4 protein is considered more informative o determine whether HPV is in the replication cycle or not. The authors should provide a rationale for not studying HPV E4 expression. The authors should also provide images of HPV E6 protein expression.

3. The authors need to provide further details of the immunohistochemistry scoring system. How did the authors determine EGFR high/low expression?

4. Did the authors attempt to stratify the smokers as 'current smokers' and 'past smokers' separately? Or to perform the analysis presented in Table 2 by stratifying the patients based on number of smoke-years instead of as 'never smokers'/'past or current smokers'?

5. Why did the authors not perform in-situ hybridization for HPV RNA or HPV DNA?

6. The authors should describe any whether any validation steps hav been planned and what these steps are in the discussion section.

Author Response

Response to Reviewer 1 Comments

Thank you for your comments and suggestions.

Point 1: The authors need to provide the detailed immunohistochemistry protocol in the supplement.

Response 1: We had added the detail immunohistochemistry protocol in the supplement. Please see the Supplementary Table 1.

Pont 2: The authors report the findings on HPV E16E6 and HPV18E6 protein expression. Although E6 is the major HPV oncoprotein, HOV E4 protein is considered more informative o determine whether HPV is in the replication cycle or not. The authors should provide a rationale for not studying HPV E4 expression. The authors should also provide images of HPV E6 protein expression.

Response 2: We had added the rational for not studying using HPV E4 expression in clinical study in the section of 2.7 Immunochemistry. The image of HPV E6 expression was shown in Figure 1.

Point 3: The authors need to provide further details of the immunohistochemistry scoring system. How did the authors determine EGFR high/low expression?

Response 3: The immunohistochemistry scoring system was according the international IHC scoring (reference 20, 21). Please see the figure on reference 11 and Figure 1.

Point 4: Did the authors attempt to stratify the smokers as 'current smokers' and 'past smokers' separately? Or to perform the analysis presented in Table 2 by stratifying the patients based on number of smoke-years instead of as 'never smokers'/'past or current smokers'?

Response 4: This study did not separate the past smokers and current smokers due to the same exposure history to nicotine, so we pooled together for further analyses. Besides, the sample size in this study was small, so we have difficulty to stratify the patients for smoking history. In the future, we might try to analyze the by stratifying the current/past smokers or the number of smoker-years.

Point 5: Why did the authors not perform in-situ hybridization for HPV RNA or HPV DNA?

Response 5: We added the reasons of selecting immunohistochemistry in blocks, not performing in-situ hybridization for HPV RNA or HPV DNA in this study. Please see section 2.7 immunohistochemistry of material and methods.

Point 6:  The authors should describe any whether any validation steps hav been planned and what these steps are in the discussion section.

Response 6: We had described the future work in the discussion (line 375-381).

Jinn-Li Wang

Taipei Medical University

Reviewer 2 Report

In this study, Wang et al. establish lung carcinoma cell lines expressing individual HPV16 oncoproteins, and investigate the effects these oncoproteins have on endogenous EGFR, and on the sensitivity to cisplatin. 

The data is straightforward, and the authors admit the limitations of their work, in the size of the study number and models to study a potential mechanism. There is evidence from other labs using cervical and head and neck models that HPV confers resistance to cisplatin treatment, and this study builds upon that work.

I would suggest that future work involves either the entire HPV genome or a combination of oncoproteins in these cell lines, so as to dissect the effects in a system more closely related to in vivo; the individual oncoproteins are not found in high levels where the viral genome is not integrated. I realise that this is outside of the scope of the current study, but may be worth discussing.

Author Response

Response to Reviewer 2 Comments

Point 1: I would suggest that future work involves either the entire HPV genome or a combination of oncoproteins in these cell lines, so as to dissect the effects in a system more closely related to in vivo; the individual oncoproteins are not found in high levels where the viral genome is not integrated. I realise that this is outside of the scope of the current study, but may be worth discussing.

Response 1:

 Thank you for your comments and suggestions. We had planned to develop patient-derived xenograft animal model (using the fresh lung adenocarcinoma cancer tisssues with or without HPV infections) and examine the related molecular experiments instead entire HPV genome or a combination of oncoprotein. I had added the future work in the discussion (line 375~381).

Jinn-Li Wang

Taipei Medical University
